# A Non-Contact Fault Diagnosis Method for Bearings and Gears Based on Generalized Matrix Norm Sparse Filtering

**DOI:** 10.3390/e23081075

**Published:** 2021-08-19

**Authors:** Huaiqian Bao, Zhaoting Shi, Jinrui Wang, Zongzhen Zhang, Guowei Zhang

**Affiliations:** 1College of Mechanical and Electronic Engineering, Shandong University of Science and Technology, Qingdao 266590, China; bhqian@sdust.edu.cn (H.B.); 201983050064@sdust.edu.cn (Z.S.); skd996576@sdust.edu.cn (Z.Z.); 201883050093@sdust.edu.cn (G.Z.); 2College of Energy and Power Engineering, Nanjing University of Aeronautics and Astronautics, Nanjing 210016, China

**Keywords:** fault diagnosis, acoustic signal, sparse filtering, generalized matrix norm, softmax

## Abstract

Fault diagnosis of mechanical equipment is mainly based on the contact measurement and analysis of vibration signals. In some special working conditions, the non-contact fault diagnosis method represented by the measurement of acoustic signals can make up for the lack of contact testing. However, its engineering application value is greatly restricted due to the low signal-to-noise ratio (SNR) of the acoustic signal. To solve this deficiency, a novel fault diagnosis method based on the generalized matrix norm sparse filtering (GMNSF) is proposed in this paper. Specially, the generalized matrix norm is introduced into the sparse filtering to seek the optimal sparse feature distribution to overcome the defect of low SNR of acoustic signals. Firstly, the collected acoustic signals are randomly overlapped to form the sample fragment data set. Then, three constraints are imposed on the multi-period data set by the GMNSF model to extract the sparse features in the sample. Finally, softmax is used to as a classifier to categorize different fault types. The diagnostic performance of the proposed method is verified by the bearing and planetary gear datasets. Results show that the GMNSF model has good feature extraction ability performance and anti-noise ability than other traditional methods.

## 1. Introduction

Rotating machine plays a crucial part in steam turbine, electric generator, and engine [1,2]. Bearings and gears, as important rotating and vulnerable parts, will inevitably fail during the operation of the equipment. Consequently, diagnosing gear and bearing faults as quickly as possible has become a hot research topic at this stage [3].

With the development of deep learning, intelligent fault diagnosis methods have achieved remarkable success [4]. Wang et al. [5] combined stacked autoencoders (SAEs) and batch normalization to achieve fast feature extraction and fault diagnosis. Li et al. [6] connected convolutional neural networks (CNN) and S-transform (ST) algorithm to enhance the characteristics of learning ability, namely, ST-CNN. Yu et al. [7] developed a three-stage semi-supervised learning-based intelligent bearing fault diagnosis method which combined data augmentation (DA) and metric learning. Zheng et al. [8] proposed a fault diagnosis method with high diagnostic accuracy based on parallel algorithms. To overcome the lack of learning features of traditional autoencoders, a standardized sparse autoencoder is developed by Jia et al. [9]. Although these methods overcome the difficulties of signal processing, they are mainly based on the measurement and analysis of vibration signals. However, in some special working conditions, the sensor installation is inconvenient and the vibration signal can not be feedback, which limits the contact fault diagnosis method represented by the measurement of vibration signal. Therefore, the non-contact fault diagnosis method represented by the measurement of acoustic signal can replace the vibration signal as the means of fault diagnosis in some special environments [10]. Zhang et al. [11] proposed a novel transfer learning method named hybrid distance-guided adversarial network (HDAN) for intelligent fault diagnosis. Wang et al. [12] added *l*1/2 norm to the objective function of SF to realize intelligent fault diagnosis under velocity fluctuation. Han et al. [13] stacked SF and normalized the input of each layer to realize bearing fault diagnosis under the condition of speed fluctuation.

Acoustic signal adopts non-contact measurement method, which is more widely used than vibration signal and has lower cost [14]. Therefore, it has become a trend to apply acoustic signal to fault diagnosis [15]. Zhang et al. [16] converted acoustic signals into geometric structure graphs and proposed a depth graph convolutional network (DGCN) method for fault diagnosis of rolling bearings. Liu et al. [17] transformed acoustic signals into spectrograms and used stacked sparse autoencoders to extract fault information. Glovacz [18] developed a simplified frequency selection method (SMOFS) based on acoustic signals for induction motor fault identification. Parvathi et al. [19] applied reasonable expansion wavelet transform (RADWT) to preprocess acoustic signals for fault diagnosis of three-phase induction motors. The above intelligent fault diagnosis technologies rely on human prior knowledge and related signal processing knowledge, which is time consuming and laborious. In addition, the low SNR of acoustic signal increases the difficulty of signal denoising and feature extraction. Therefore, improving the ability of fault feature extraction is the key problem of acoustic signal based intelligent fault diagnosis technology.

In the process of feature extraction [20], we can obtain additional useful information by learning sparse features. Sparsity means the output items are mostly zero to ensure that excessive information features will not be lost [21]. Some well-known sparsity feature learning methods are sparse restricted boltzmann machine (RBM) [22], sparse autoencoders [23], and sparse filtering (SF) [24]. Sparse coding learns many overly comprehensive bases [25]. Therefore, the sparse coefficient can be used to represent the sample. Similar to sparse autoencoders, sparse RBM can be realized after considering penalty with sparse terms. Considering that sparse coding and sparse RBM [22] have sparse penalty terms, sparse features can be obtained. SF is an efficient and simple unsupervised feature learning algorithm [26]. It can bypass the estimation of data distribution and only requires optimizing a simple cost function to directly optimize the feature distribution. Despite the SF can achieve ideal diagnosis result, the input dataset are all vibration signals. It is still difficult to obtain so high accuracy through the acoustic signal.

In order to overcome the above shortcoming, a novel non-contact fault diagnosis method based on the generalized matrix norm sparse filtering (GMNSF) is proposed in this paper. In this method, the generalized matrix norm is introduced into the sparse filtering to seek the optimal sparse feature distribution to overcome the defect of low SNR of acoustic signals. First of all, the acoustic signals collected are randomly overlapping to form the sample fragment data set. Then, three constraints are imposed on the multi-period data set by the generalized matrix norm sparse filtering model to extract the sparse features in the sample. Finally, Softmax classifier is used to output bearing health states corresponding to different acoustic signals.

The remainder of this paper is organized as follows. Section 2 particularly introduces the proposed GMNSF method in fault diagnosis. Section 3 investigates two diagnosis cases of planetary gearbox and rolling bearing datasets to verify the accuracy of GMNSF method. Furthermore, the optimal parameters are assigned through comparative analysis. Section 4 analyzes the weight matrix. The last section comes the conclusions.

## 2. Proposed Method

### 2.1. GMNSF

As shown in Figure 1, the structure of GMNSF is a two-layer network. The inputs and outputs of GMNSF are the collected samples and learned features. For the training dataset ${\left\{x^{i}\right\}}_{i=1}^{M}$, where M is the number of samples and $x^{i}\in {ℜ}^{N*1}$ is a sample with N input variables, the feature form ${\left\{f_{j}^{i}\right\}}_{i=1}^{M}$ can be obtained by weight matrix W, where each row of ***W*** is a filter applied to each sample. Concretely, let $f_{j}^{i}$ denote the *j*th feature value (row) for the *i*th example (column). g(.) is used as activation function. The feature calculation is
(1)$$f_{j}^{i}=g(W_{j}x^{i})$$

In practice, the activation function g(.) can be arbitrarily selected. To overcome the disadvantage that the cost function is not smooth, the soft absolute function $g(x)=\sqrt{x^{2}+\varepsilon }$ can be used as the activation function, where ε is an extremely small parameter, and $\varepsilon =10^{-8}$.

$f_{j}^{i}$ is the feature value of the *i*th row and *j*th column in characteristic matrix, and $f_{j}^{i}$ is normalized to an equal activation value, so that each feature is divided by its $l_{2}$-norm in all samples.
(2)$${\tilde{f}}_{j}=\frac{f_{j}}{{\left\|f_{j}\right\|}_{2}}$$
where $f_{j}$ denotes each column of the matrix, and ${\tilde{f}}_{j}$ denotes the normalized features.

The features of each sample are normalized so that the features of all samples fall on the unit sphere of $l_{2}$-norm.
(3)$${\hat{f}}^{i}=\frac{{\tilde{f}}^{i}}{{\left\|f^{i}\right\|}_{2}}$$
where $f^{i}$ denotes each row of the matrix. and ${\hat{f}}^{i}$ denotes the normalized features.

Then, the matrix norm penalty is used for sparsity constrained optimization. Suppose a data set has M samples, the objective equation is
(4)$$C={\left[\sum_{i=1}^{M}{\left(\sum_{j=1}^{L}{\left(\frac{{\tilde{f}}_{j}^{i}}{{\left\|{\tilde{f}}_{j}^{i}\right\|}_{2}}\right)}^{r}\right)}^{\frac{s}{r}}\right]}^{\frac{1}{s}}$$
where r and s are matrix norms for measuring standardized features, *L* denotes the sample dimension.

The generalized matrix norm can ensure that all the inactive eigenvalues will be approximately uniformly distributed in the eigenspace. In the feature matrix, the lifetime sparsity of the feature has been optimized in the optimization process of population sparsity and high dispersity.

Figure 2 shows the sparse feature learning process of GMNSF. The learned eigenvector can be expressed as $f^{1}={\left[f_{1}^{1}f_{2}^{1}\right]}^{T}$ and $f^{2}={\left[f_{1}^{2}f_{2}^{2}\right]}^{T}$. In the training process of GMNSF, both samples must be mapped to the unit circle of l2-norm, and the normalized characteristics are limited to a unit circle.

In this paper, the cost function C is minimized by *L-BFGS* algorithm. In particular, the gradient of cost function for W is
(5)$$\frac{\partial C}{\partial W}=\left(\frac{\partial C}{\partial f}\cdot \frac{Wx}{\sqrt{{\left(Wx\right)}^{2}+\varepsilon }}\right)\cdot x^{T}$$
(6)∂C∂fji=∂C∂f˜ji⋅1∑j=1Lfji212−fji∑j=1Lfji232∑j=1Lfji⋅∂C∂f˜ji
(7)∂C∂f˜li=∑i=1M∑j=1Lf^jirsr1s−1⋅∑j=1Lf^jirsr−1⋅f^jir−1

### 2.2. Intelligent Fault Diagnosis Framework

As shown in Figure 3, the intelligent fault diagnosis method in this paper is divided into three stages: model training, feature extraction, and fault recognition. First, the GMNSF model is trained to obtain the optimized W through the acoustic dataset. Second, the optimized W is used to extract features. Finally, softmax regression is empolyed to classify the health status according to the learned features.

Model training: The GMNSF model is trained through the original acoustic dataset. Set the training sample set as ${\left\{x^{i},y^{i}\right\}}_{i=1}^{M}$where M is the number of samples, $x^{i}\in {ℜ}^{N\times 1}$ denotes the N input variables in the i sample, and $y^{i}$ is the label of the sample $x^{i}$. The sample is randomly divided into $N_{s}$ segments and each segment includes $N_{in}$ input variables. Thus, a training set ${\left\{s^{j}\right\}}_{j=1}^{MN_{s}}$ contains $MN_{s}$ segments, where $s^{j}\in {ℜ}^{N_{in}\times 1}$ is the jth segment of the training dataset. Noticeably, the segments are obtained through overlapping. The following are the specific steps of model training. Then, the obtained ${\left\{s^{j}\right\}}_{j=1}^{MN_{s}}$ is directly inputted to GMNSF to train the weight matrix ***W***.Feature extraction: The discriminant features can be obtained by the optimal weight matrix W. The training sample $x_{i}$ is separated into K sections, and the K is an integer and equal to $N/N_{in}$. The dataset ${\left\{x_{k}^{i}\right\}}_{k=1}^{K}$ is composed of K segments and $x_{k}^{i}\in {ℜ}^{N_{in}\times 1}$. Then, the trained sparse filtering model is used to calculate the local feature ${\overset{⌣}{f}}_{k}^{i}\in {ℜ}^{N_{out}\times 1}$ of each sample. Learned features are obtained by combining local features using the average method, leading to
(8)$${\overset{⌣}{f}}^{i}=\frac{1}{K}\sum_{k=1}^{K}{\overset{⌣}{f}}_{k}^{i}$$Fault recognition: The learned features of all samples are combined with the labels and then trained through the softmax classifier. First, Z-Score is normalized to activate training and test data. The calculation method is as follows:
(9)$$F=\frac{f-\bar{f}}{\sigma }$$where $\bar{f}$ denotes the mean value of f, and σ represents the standard deviation. Thus, the rescaled matrix f owns the properties of a standard normal distribution.Then, the softmax regression model is trained by the learned characteristic set ${\left\{{\overset{⌣}{f}}^{i}\right\}}_{i=1}^{M}$ and the healthy condition label set ${\left\{y^{i}\right\}}_{i=1}^{M}$, $y^{i}\in \left\{1,2,\ldots ,D\right\}$. The softmax regression output probability $\rho \left(y^{i}=d/{\overset{⌣}{f}}^{i}\right)$ that ${\overset{⌣}{f}}^{i}$ is the label of the feature vector. The softmax regression model of weight matrix θ is acquired by minimizing the nest cost function.
(10)$$H(\theta )=\frac{\lambda }{2}\sum_{d=1}^{D}\sum_{j=1}^{{Ν}_{ο\upsilon \tau }}{\left(\theta_{i}^{j}\right)}^{2}-\frac{1}{Μ}\left[\sum_{i=1}^{M}\sum_{d=1}^{D}\left\{y^{i}=d\right\}\log\frac{e^{\theta_{d}}{\overset{⌣}{f}}^{i}}{\sum_{c=1}^{D}e^{\theta_{c}}{\overset{⌣}{f}}^{i}}\right]$$
where 1{.} is the indicator function, *M* denotes the training sample size, and *D* denotes the category number. λ is the parameter of the weight decay term.

## 3. Experimental Validation

Rolling bearings and gears are the major components of rotating machinery, and their health conditions are closely related to the normal operation of the equipment. In this paper, the rolling bearing data set and planetary gear data set are taken as the research objects to evaluate the effect of the proposed method.

### 3.1. Rolling Bearing Data Verification and Analysis

The bearing data set is from Shandong University of Science and Technology (SDUST). Experimental equipment includes motor, coupling, planetary gearbox, bearing seat, and turntable. Figure 4a shows the main components. Acoustic sensor is placed 30 cm from the bearing housing. The acoustic signals of the bearing under the stable condition are collected by the LMS data acquisition instrument, and the sampling frequency is set at 25.6 kHz. The bearing model is an N205EU cylindrical roller bearing. The dataset contains 9 kinds of health conditions in Figure 4b: inner race fault 0.2 mm, and 0.4 mm (IF0.2 and IF0.4), roller fault 0.2 mm and 0.4 mm (RF0.2 and RF0.4), outer race fault 0.2 mm and 0.4 mm (OF0.2 and OF0.4), roller and outer race fault 0.2 mm and 0.4 mm (ROF0.2 and ROF0.4), and normal condition (NC). 200 samples are collected for each health state, so the bearing data set contains a total of 1800 samples, among which each sample contains 1200 time domain points. The samples used for training accounts for 50%, and the rest are selected for testing. Other parameters are randomly set as follows: the number of overlapping segments is set as 50, $N_{s}=50$. The output dimension is equal to the segment length, $N_{in}=N_{out}=100$. To eliminate the random effect, each set of experiments is conducted 25 times.

Figure 5 shows the effect of matrix norm on sparse performance. The parameters of the matrix norm *r* and *s* are set between 1 and 10. As can be seen from Figure 5, the accuracy of the model drops rapidly when r=2. According to Equation (4), the algorithm constraint fails at that time, which leads to the decrease of accuracy. At the same time, it can be found that the accuracy of the model is related to the ratio. The closer the ratio is to 1, the lower the accuracy is, and the accuracy gradually increases with the increase of the ratio difference. In addition, the accuracy reached the highest when *r* = 8 and then gradually decreased. According to the above analysis, *r* = 8 and *s* = 1 are the best and the average accuracy is 97.7%.

The input and output dimensions of the model as well as the size of the data set have a great impact on the algorithm. In order to study the performance of GMNSF under different parameter settings, Figure 6 shows the impact of the input dimension on diagnosis performance. It can be seen that the average accuracy increases with the increase of the input dimension when the input dimension is less than 150; the accuracy does not change significantly when the input dimension is greater than 150 and the accuracy decreases significantly when the input dimension is greater than 300. The experiment shows that the bigger the input dimension is not the better. Considering the calculation accuracy and time, the best input dimension output in this experiment is 150.

Figure 7 illustrates the influence of output dimension on diagnosis performance. It is clearly to seen that the average calculation time of the algorithm increases with the increase of the output dimension, but the calculation accuracy does not change significantly when the output dimension is greater than 150. Therefore, the experiment shows that the bigger the output dimension is not the better. Considering the accuracy and calculation efficiency, the optimal output dimension in this experiment is 150.

Figure 8 shows the impact of the number of segments on diagnosis performance. It is observed that the test accuracy gradually increases when the number of sections is less than 50, and the calculation time slowly increases. In addition, the increasing trend of test accuracy becomes slow, and the calculation time increases sharply when the number of sections is greater than 50. In conclusion, $N_{S}=50$ is selected for sample segmentation training.

On the basis of the above-mentioned analysis, the average accuracy of the proposed method on bearing dataset is 97.9%, the error is 0.96%, and the average calculation time is 12 s. In addition, a confusion matrix is used to display the classification result as shown in Figure 9. In the figure, the abscissa is the predicted label of the test sample, and the ordinate is the true label of the test sample. The diagonal of the matrix is the correct classification accuracy rate, and the rest is the error classification rate. The results show that each health condition can be distinguished, and the classification accuracy is above 97.93%. Specially, the testing accuracy rate to NC, IF0.4, RF0.2, and OF0.4 is 100%. In addition, 6 samples of IF0.2, 1 sample of RF0.4, 7 samples of OF0.2, 1 sample of ROF0.2, and 5 samples of ROF0.4 are misclassified as other health conditions by GMNSF.

To better demonstrate the advantage of GMNSF, we make a comprehensive comparison with several existing methods as shown in Table 1. The first method, standard sparse filtering (SF) [27] method can achieve the accuracy of 92.96 ± 1.22%. In the second method, a fast intrinsic component filtering (ICF) [28] and 87.35 ± 1.58% testing accuracy are obtained. In the third method, a feature extraction algorithm based on convolutional sparse filter (CSF) [29], which can obtain 73.35 ± 3.78%. In the fourth method, the accuracy of the developed GMNSF method is 97.93 ± 0.96%, which is better than all comparison methods.

### 3.2. Gear Data Verification and Analysis

In order to simulate the actual working conditions, acoustic data are collected in the gear speed random fluctuation state, so as to verify the diagnostic performance of the proposed generalized matrix norm sparse filtering method. The experimental data in this section are collected by the planetary gearbox failure test bench. In order to show the speed of the collected sound signal, the speed fluctuation diagram under five health conditions is shown in Figure 10. It can be seen that the speed fluctuation range is between 500 r/min and 3000 r/min. The irregular speed fluctuation in a large range makes the extracted signal features closer to the actual working conditions. Figure 11 shows 5 health conditions: normal condition (NC), planetary wheel fracture (PF), planetary wheel wear (PW), sun wheel fracture (SF), and sun wheel pitting (SP). The sampling frequency is 25.6 kHz, 200 samples are collected under each gear health condition, and each sample contains 1200 input variables. Figure 12 shows the time domain signals of the five types of gear health condition. The methods based on time domain have difficulty distinguishing the health condition of gears. Therefore, these health conditions can be distinguished by the proposed method. The parameters are set to $r=8,s=1,N_{in}=150,N_{out}=150,N_{s}=50$; 50% of samples are used for model training; and the rest are selected for performance diagnosis. A total of 1000 segments are trained. The average testing accuracy is 96.6%, standard deviation is 1.43%.

Additional diagnostic information can be seen in Figure 13. It can be observed that compared with other classical methods, GMNSF still has the best fault diagnosis performance, with a maximum test accuracy of 96.6±1.43%. The diagnostic results of SF and CSF were 95.6±1.45% and 89.24±3.42%, respectively. Therefore, the proposed GMNSF method also shows a good anti-noise ability when processing the acoustic signals of the gearbox.

To further describe the differentiation of learning features by the proposed method, T-SNE [30] is used to reduce the dimension of high-dimensional features to achieve visualization. Figure 14a shows the dimension reduction result of GMNSF. Apparently, GMNSF can perfectly divide these feature vectors into five parts. The proposed method is compared with the traditional SF, and Figure 14b presents the dimension reduction result of SF. It is easy to find that some overlaps exist among different categories. Therefore, the proposed method can obtain a better clustering performance than traditional SF.

## 4. Discuss Weight Matrix

To further illustrate the sparse feature extraction performance of GMNSF. The standard SF is compared with the proposed GMNSF feature vector by using the rolling bearing data set, as shown in Figure 15. It is observed that the feature distribution obtained by GMNSF has obvious sparsity and strong filtering performance, and only a few feature points have non-zero values. However, the feature vector sparsity of SF is poor. This indicates that GMNSF can extract more sparse features than SF.

In addition, 9 row vectors of weight matrix trained by GMNSF are randomly selected as shown in Figure 16a. Figure 16b presents the spectra acquired by the fast Fourier transform (FFT). The result shows that the weight vector trained by GMNSF model presents an oscillation curve in the time domain graph, and the frequency band of the spectrum is narrow. These weight vectors have the same time-frequency characteristics as the Gabor filters used as signal bandpass filters [31]. Thus, each weight vector can be considered a Gabor filter. Remarkably, most of the weighted part of the spectra will suppress different center frequencies, indicating that the proposed method has a good band-pass basis for acoustic signals. As shown in Figure 17, the row vector trained by the traditional SF method has obvious components and no obvious peak value in other frequency ranges, so SF does not show sufficient feature extraction capability.

## 5. Conclusions

The current study proposes a generalized matrix norm sparse filtering to improve the feature learning ability of raw sparse filtering. The generalized matrix norm is introduced into the standard sparse filtering to obtain the optimal sparse distribution. The diagnosis results of rolling bearing dataset and planetary gearbox dataset show that GMNSF model has higher test accuracy and more stable weight quality for acoustic signals. In addition, the selection of sparsity parameter is beneficial in improving diagnostic performance, and the diagnostic accuracy of the proposed method is better than that of existing methods.

However, the hyperparameters of the proposed method still need to be manually adjusted. The automatic selection of the parameters may make GMNSF easier to be used in practice. Therefore, the next step is to adaptively adjust these parameters by some optimization techniques, so as to improve the intelligence of the model and raise its performance.

## Figures and Tables

**Figure 1 entropy-23-01075-f001:**
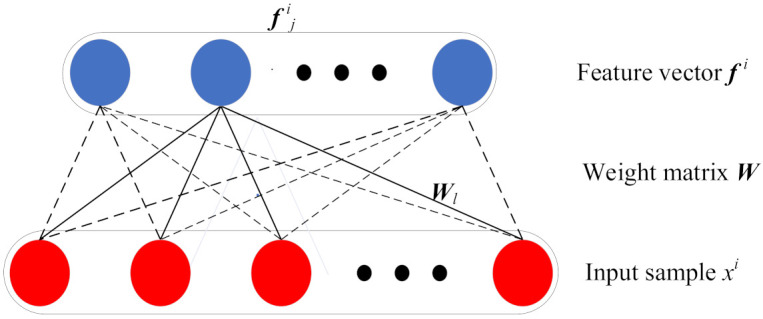
Network structure of GMNSF.

**Figure 2 entropy-23-01075-f002:**
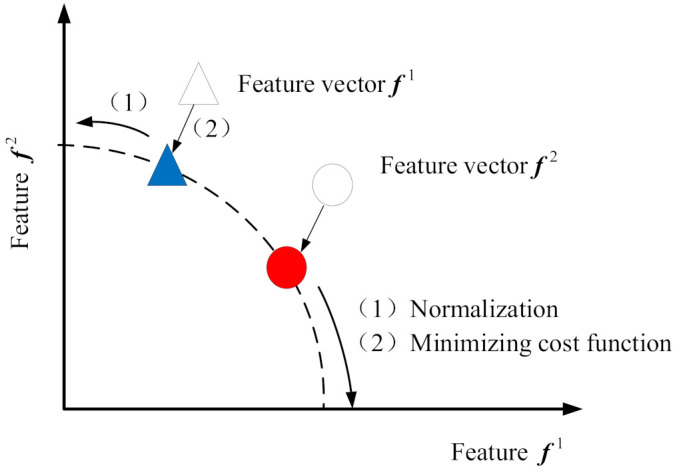
Sparse feature learning process of GMNSF.

**Figure 3 entropy-23-01075-f003:**
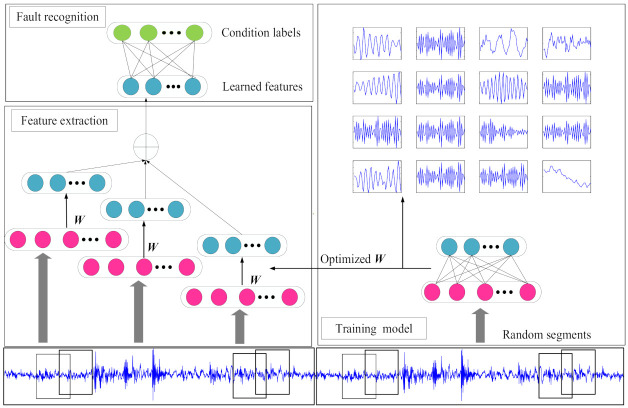
Intelligent fault diagnosis framework based on GMNSF.

**Figure 4 entropy-23-01075-f004:**
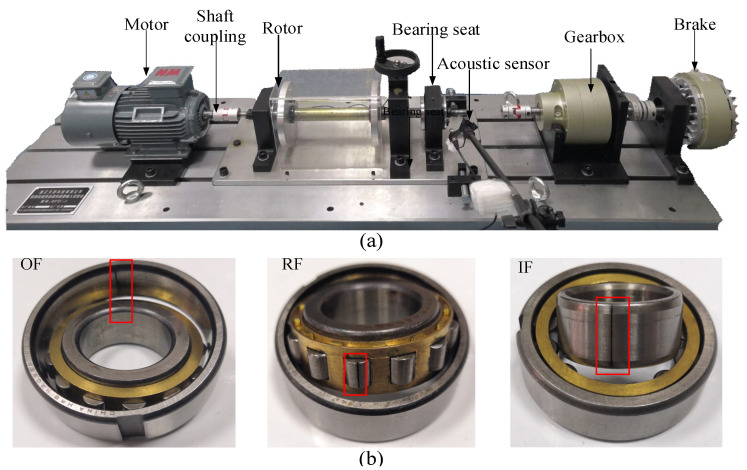
(**a**) Rotor test bench and (**b**) Bearing fault feature diagram.

**Figure 5 entropy-23-01075-f005:**
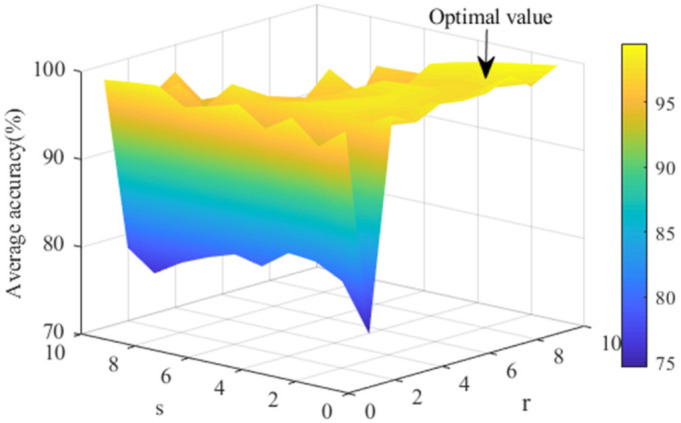
Diagram of different matrix norm test results.

**Figure 6 entropy-23-01075-f006:**
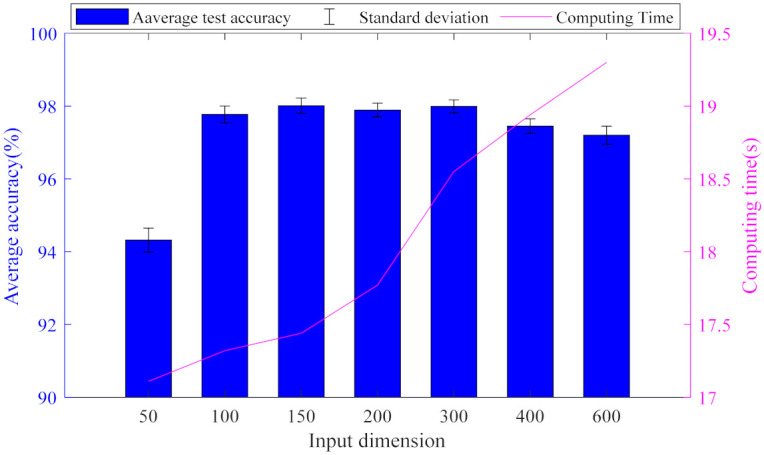
Diagram of test results for different input dimensions $N_{in}$.

**Figure 7 entropy-23-01075-f007:**
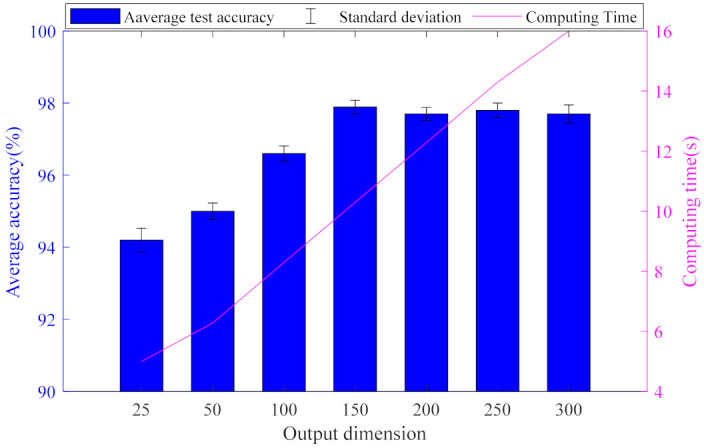
Diagram of test results for different output dimensions $N_{out}$.

**Figure 8 entropy-23-01075-f008:**
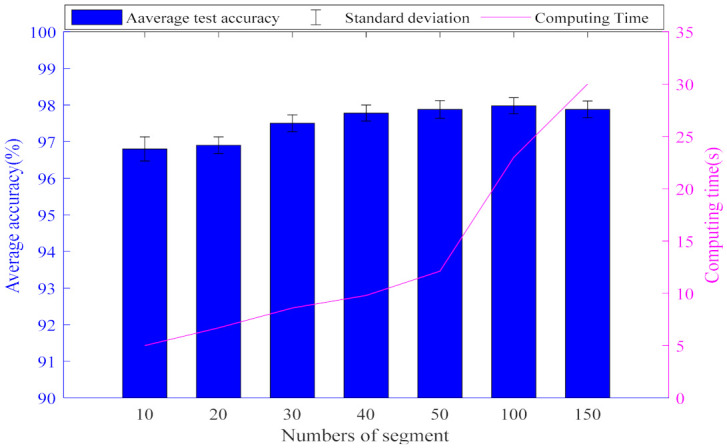
Diagram of test results for different numbers of sections *Ns.*

**Figure 9 entropy-23-01075-f009:**
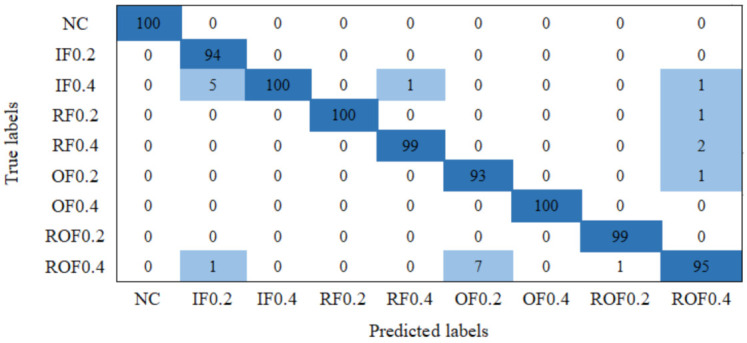
Confusion matrix of the bearing dataset.

**Figure 10 entropy-23-01075-f010:**
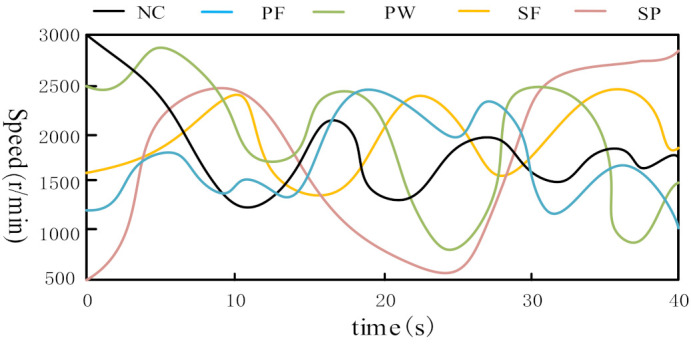
Gear speed fluctuation diagram.

**Figure 11 entropy-23-01075-f011:**
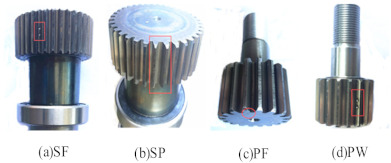
Gear fault feature diagrams.

**Figure 12 entropy-23-01075-f012:**
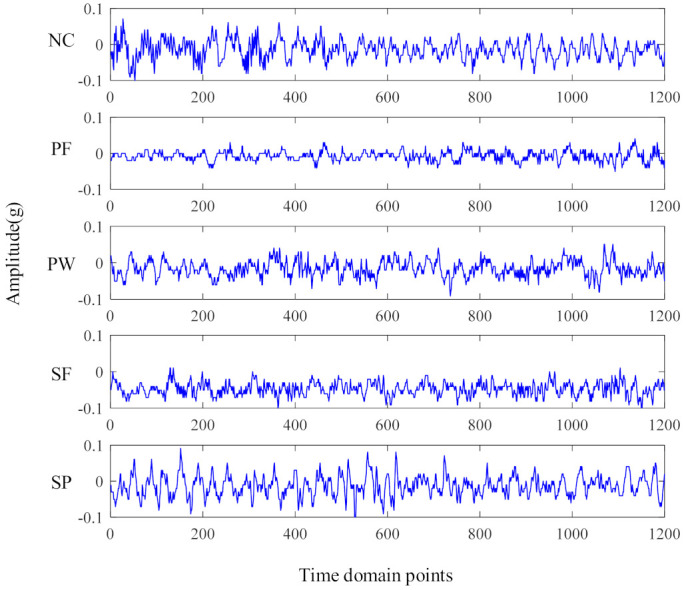
Time domain waveforms of gear health condition.

**Figure 13 entropy-23-01075-f013:**
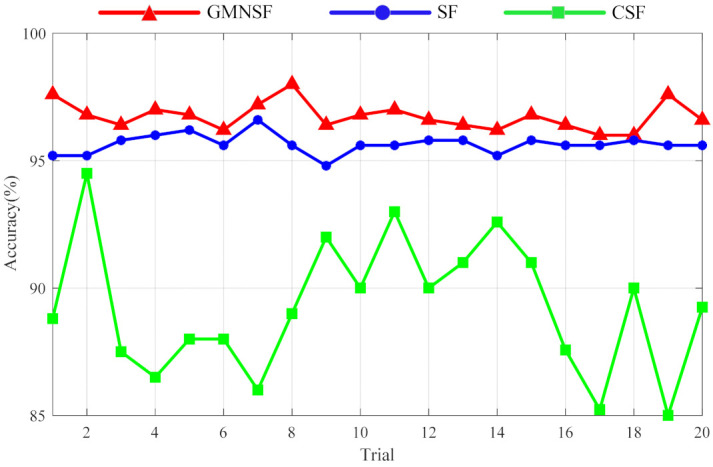
Comparison of testing accuracy of gear data sets.

**Figure 14 entropy-23-01075-f014:**
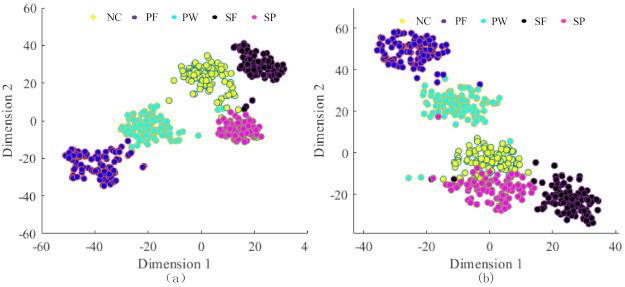
Dimension reduction results: (**a**) GMNSF, (**b**) traditional sparse filtering.

**Figure 15 entropy-23-01075-f015:**
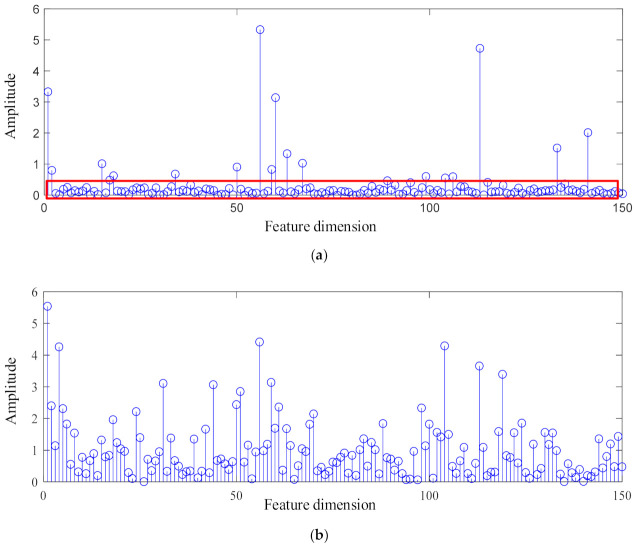
Learned feature vectors of NC condition: (**a**) GMNSF, (**b**) SF.

**Figure 16 entropy-23-01075-f016:**
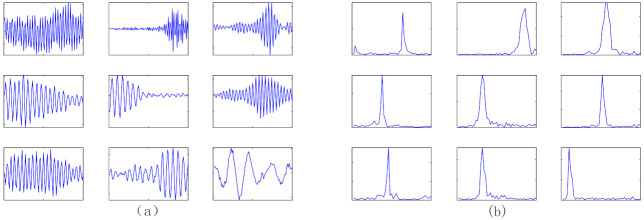
The weight vector learned by GMNSF model: (**a**) time domain, (**b**) Frequency domain.

**Figure 17 entropy-23-01075-f017:**
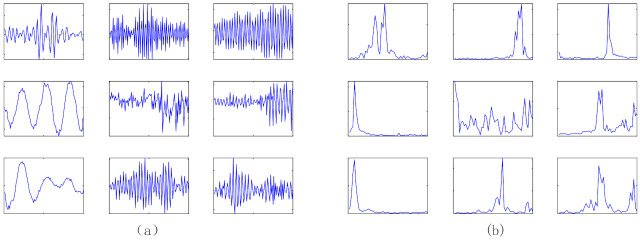
The weight vectort learned by SF model: (**a**) time domain, (**b**) Frequency domain.

**Table 1 entropy-23-01075-t001:** Performance comparison between the proposed GMNSF method and some classical methods.

Method	Description	Training Samples (%)	Testing Accuracy (%)
1	SF	50	92.96 ± 1.22%
2	ICF	50	87.35 ± 1.58%
4	CSF	50	73.35 ± 3.78%
5	GMNSF	50	97.93 ± 0.96%

## Data Availability

The data used to support the findings of this study are available from the corresponding author upon request.

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
