# Peer review of "A Non-Contact Fault Diagnosis Method for Bearings and Gears Based on Generalized Matrix Norm Sparse Filtering"

_entropy, 2021, doi:10.3390/e23081075_

Round 1
Reviewer 1 Report
The paper is interesting. I have some special concerns. If they can be considered when revising the paper, I am glad to recommend the publication of its version.
Comment 1: The literature review part needs to be improved. Authors need to add new literature reviews focusing on the intelligent fault diagnosis in the Introduction.
Comment 2: Following comment 1, there are Data-driven fault diagnosis for traction systems in high-speed trains: A survey, challenges, and perspectives and A review of intelligent fault diagnosis for high-speed trains: Qualitative approaches. Entropy.
Comment 3: In Subsection 3.1, the authors only collected the bearing data set containing a total of 1800 samples to test the validity of the proposed method, is it too little for the operation of AI methods?
Comment 4: How does the author choose the hyperparameters in the model?
Comment 5: The text given in the figures should be adjusted.
Comment 6: Please add some sentences about future analysis.
Comment 7: Even if the English writing is good, there are some minor changes needed in the vision.
Reviewer 2 Report
This paper presents a fault diagnosis method based on the generalized matrix norm sparse filtering. The generalized matrix norm is introduced into the sparse filtering to seek the optimal sparse feature distribution to overcome the defect of low signal-to-noise ratio of acoustic signals.
Description of the proposed GMNSF is poor. In the paragraph above Figure 1, what are the difference between “samples” and “data points”? Some clarifications should be given. Also “eigenvector” of which matrix is referred to here? The use of variable is a bit confusing. Here f_{i}, f^{i}, and f_{i}^{j} are used and their difference is not clear.
In this part, f is given by Eq(1) and later Eq(5), but f is also mentioned as eigenvector. How could the f given by Eq(1) or Eq(5) be eigenvector?
It is not clear how fault diagnosis is carried out.
Round 2
Reviewer 1 Report
The reviewer appreciates the revision from all the authors. Now my suggestion is to Accept.
Author Response
We sincerely thank the reviewer for your valuable comments and insights which have helped to considerably improve the quality of our previously submitted manuscript.
Reviewer 2 Report
The authors have adequately addressed my comments. From the authors’ response, it seems they use “data points” to “input variables”. It would be better if “input variables” is used instead of “data points”.
Author Response
请参阅附件。
